# GRAPH REGULARIZED ENCODER TRAINING FOR EXTREME CLASSIFICATION

## ABSTRACT

Deep extreme classification (XC) aims to train an encoder and label classifiers to tag a data point with the most relevant subset of labels from a very large universe of labels. XC applications in ranking, recommendation and tagging routinely encounter tail labels, for which the amount of training data is exceedingly small. One way to tackle the tail label problem is to use additional data - often structured as a graph associated with documents and labels - graph metadata. Graph Convolutional Networks (GCNs) present a convenient but computationally expensive way to leverage this graph metadata and enhance model accuracies in these settings. However, GCNs struggle to make predictions for a novel test point when it has no edge in the graph. The paper notices that in these settings, it is much more effective to use graph data to regularize encoder training than to implement a GCN. Based on these insights, an alternative paradigm RAMEN is presented to utilize graph metadata in XC settings that offers a significant performance boost with zero increase in inference computational costs. RAMEN scales to datasets with millions of labels and offers prediction accuracy up to 15% higher on benchmark datasets than state of the art methods, including those that use graph metadata to train GCNs. RAMEN also offers 10% higher accuracy over the best baseline on a proprietary recommendation dataset sourced from click logs of a popular search engine. Code for RAMEN will be released publicly upon acceptance.

## 1 INTRODUCTION

Extreme classification (XC) refers to a supervised machine learning paradigm where multi-label learning must be performed on extremely large label spaces. Thus, a data point must be annotated with a subset of labels most relevant to it. The ability of XC to handle enormous label sets with millions of labels makes it an attractive choice for applications such as product recommendation (Medini et al., 2019; Dahiya et al., 2021b; Mittal et al., 2022; Kharbanda et al., 2022), document tagging (Babbar & Schölkopf, 2017; You et al., 2019; Chang et al., 2020), search & advertisement (Prabhu et al., 2018b; Dahiya et al., 2021b; Jain et al., 2016), and query recommendation (Jain et al., 2019; Chang et al., 2020). The key appeal of XC comes from the prospect of accurately tagging rare/tail labels relevant to a data point. Recommendations for rare but relevant objects can meaningfully improve user experience and the ability to associate rare tags with objects such as web documents can offer fine-grained object descriptions. A label is called tail if very few training data points are tagged with that label. XC applications can exhibit extreme label skew and more than 75% of the labels could appear in fewer than 10 training points (Jain et al., 2016; Dean, 2020). The tail problem is further aggravated due to missing labels since tail labels are also at higher risk of going missing (Jain et al., 2016). In solving the tail-data problem, XC approaches rely on metadata. Beyond textual label descriptions (Mittal et al., 2021a; Dahiya et al., 2021a; 2023), the auxiliary metadata can augment the meagre supervision available for tail labels and is typically available in the form of multi-modal descriptions such as images (Mittal et al., 2022), or graphs (Mittal et al., 2021b; Saini et al., 2021). In this paper, we focus on graph data which can be inferred in several applications, *e.g.,* hyperlink graphs for document tagging and queries co-occurring in the same search session for ad placement.

**Graph metadata in XC:** Graph metadata has been used in XC to (a) enhance item representations, and (b) handle missing labels. Examples of the former include OAK (Mohan et al., 2015), Graph-Former (Yang et al., 2021), and PINA (Chien et al., 2023) which use textual descriptions of an item along with graph metadata to learn item embeddings via graph convolutional networks (GCN). These

Figure 1: RAMEN uses graph metadata to regularize encoder during training and unlike GCN RAMEN requires no additional inference cost. (a) RAMEN training uses graph metadata to regularize the encoder $\mathcal{E}_\theta$. (b) RAMEN's encoder requires no additional information to compute an accurate representation of the test point. (c) In GCNs, inference is a computationally expensive two stage pipeline, where the test point is first embedded in the graph and then the linked nodes are used to compute the final representation. RAMEN can be 2× faster, and 3-4% more accurate, than GCNs.

algorithms rely on a two stage retrieval pipeline wherein, for a novel test point, graph metadata nodes are first retrieved and a GCN combines them with the test point. The new representation is then used in second stage to retrieve the relevant labels. Graphs traversal can also help discover missing labels associated with documents. For example, consider LF-WikiSeeAlsoTitles-320K where the task is to predict related Wikipedia documents. A hyperlink graph is available which connects two Wikipedia articles with an edge if one of them contains a hyperlink to the other. A snapshot of the dataset in Figure 2 shows how a missing label "Crown group" can be recovered for the Wikipedia article "Cladistics" by traversing the graph.

**Limitations of GCN Methods**: Graphs can also be misleading in terms of linkages, and GCN's implementation posses limited applicability for real-world application. As an example of noisy linkages in graphs, consider the LF-WikiSeeAlsoTitles-320K hyperlink graph. Traversal over the graph can also lead to irrelevant labels such as "Vestigial organs" and extracting meaningful information from such noisy graphs is a challenge. Although the use of textual and graph metadata can offer enhanced model accuracy in XC and recommendation settings (Mittal et al., 2021b; Saini et al., 2021; Yang et al., 2021; Chien et al., 2023), the use of GCN architectures makes both training and inference more expensive (table 3). XC training is made challenging by the sheer size of training sets often containing millions of data points and labels, necessitating some form of negative sampling (Mikolov et al., 2013; Guo et al., 2019; Rawat et al., 2021; Reddi et al., 2018; Xiong et al., 2021). On the other hand, most XC applications demand real-time inference *i.e.*, the set of labels relevant to a test data point must be identified within milliseconds. GCNs require the (bulky) graph to be preserved at inference time to embed a test data point which increases inference time and makes deployment challenging. This paper addresses the limitations of using graph metadata in XC. Our primary research question is: ***How do we leverage graph metadata to perform accurate prediction for rare labels with zero increase in inference time?***

## 1.1 OUR CONTRIBUTIONS

To address the above question, we propose gRaph regulArized encoder training for extreME classificatioN (**RAMEN**). RAMEN is a method to effectively utilize graph metadata at scale with minimal overheads in training cost and zero overhead in model size or inference time (Table 3). RAMEN can be incorporated into existing XC systems in a modular manner with few alterations (Table 9). The key insights leading to RAMEN include a formal proof (cf. Theorm 1) that (a) in several use cases, GCN layers can be approximated by (much cheaper) non-GCN architectures and, (b) it is more effective to use graph data to regularize encoder training than it is to implement a GCN. RAMEN can handle multiple graphs – graphs over data points, graphs over labels, or both – and offers increased prediction accuracy, even when presented with noisy graphs (Section 4). While the RAMEN encoder is trained using the metadata graph, during inference, unlike baseline GCNs, RAMEN does not require graph traversal, significantly improving latency (cf. Figure 1). RAMEN scales to datasets with up to 360M labels and can offer up to 15% higher prediction accuracies over state-of-the-art methods including those that use graph metadata to train GCN. Code for RAMEN will be released publicly.

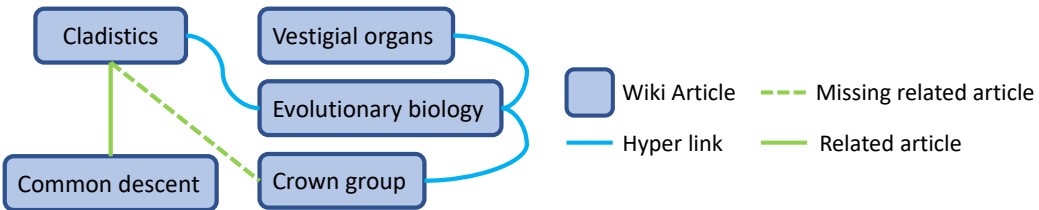

Figure 2: A snapshot from LF-WikiSeeAlsoTitles-320K dataset for the article on "Cladistics." The related article "Common descent" is tagged but the ground truth is missing the label "Crown group". Traversal on the hyperlink edges can help discover missing labels but can also lead to irrelevant labels such as "Vestigial organs".

## 2 RELATED WORK

Extreme classification (XC) is a key paradigm in several areas such as ranking and recommendation. The literature on XC methods is vast (Medini et al., 2019; Dahiya et al., 2021b; Babbar & Schölkopf, 2017; You et al., 2019; Prabhu et al., 2018b; Jain et al., 2016; 2019; Guo et al., 2019; Mittal et al., 2021a;b; Saini et al., 2021; Wydmuch et al., 2018; Zhang et al., 2018; Liu et al., 2017; Jiang et al., 2021; Chalkidis et al., 2019; Ye et al., 2020; Zhang et al., 2021; Mineiro & Karampatziakis, 2015; Jasinska et al., 2016; Khandagale et al., 2020; Tagami, 2017; Yen et al., 2017; Wei et al., 2019; Siblini et al., 2018; Barezi et al., 2019; Gupta et al., 2019; 2023). Early XC methods used fixed (bag-of-words) (Babbar & Schölkopf, 2017; Prabhu et al., 2018b; Jain et al., 2016; Mineiro & Karampatziakis, 2015; Jasinska et al., 2016; Khandagale et al., 2020; Tagami, 2017; Yen et al., 2017; Wei et al., 2019; Siblini et al., 2018; Barezi et al., 2019) or pre-trained (Jain et al., 2019) features and focused on learning only a classifier architecture. Recent advances have demonstrated significant gains by using task-specific features obtained from a variety of deep encoders such as bag-of-embeddings (Dahiya et al., 2021b; 2023), CNNs (Liu et al., 2017), LSTMs (You et al., 2019), and transformers (Jiang et al., 2021; Chalkidis et al., 2019; Ye et al., 2020; Zhang et al., 2021). Training is scaled to millions of labels and training points (Dahiya et al., 2021b) by performing encoder pre-training followed by classifier training. A data point is trained only on its relevant labels (that are usually few in number) and a select few irrelevant labels deemed most informative using negative mining (Mikolov et al., 2013; Guo et al., 2019; Xiong et al., 2021; Dahiya et al., 2021a; 2023; Faghri et al., 2018; Chen et al., 2020; He et al., 2020a; Karpukhin et al., 2020; Lee et al., 2019; Luan et al., 2020; Hofstätter et al., 2021; Qu et al., 2021).

**Label Metadata in XC**: Most XC methods use textual representation as label metadata since they allow scalable training and inference and allow leveraging good-quality pre-trained deep encoders such as RoBERTa (Liu et al., 2019b), DistilBERT base (Sanh et al., 2019), etc. Examples include encoder-only models such as DEXML (Gupta et al., 2024), TwinBERT (Lu et al., 2020) and ANCE (Xiong et al., 2021) and encoder+classifier architectures such as DECAF (Mittal et al., 2021a), SiameseXML (Dahiya et al., 2021a), X-Transformer (Chang et al., 2019), XR-Transformer (Chang et al., 2020), LightXML (Jiang et al., 2021), ELIAS (Zhang et al., 2021) and others (Ye et al., 2020; Liu et al., 2019a; You et al., 2019; Chalkidis et al., 2019). There is far less literature on the use of other forms of label metadata. For instance, ECLARE (Mittal et al., 2021b) and GalaxC (Saini et al., 2021) use graph convolutional networks whereas MUFIN (Mittal et al., 2022) explores multi-modal label metadata in the form of textual and visual descriptors for labels.

**Graph Neural Networks in Related Areas**: A sizeable body of work exists on using graph neural networks such as graph convolutional networks (GCN) for recommendation (Yang et al., 2021; Mohan et al., 2015; Hamilton et al., 2018; Chen et al., 2018; Zou et al., 2019; Huang et al., 2018; Chiang et al., 2019; Zeng et al., 2020; Zhu et al., 2021; He et al., 2020b; Yang et al., 2022). Certain methods e.g., FastGCN (Chen et al., 2018), KGCL (Yang et al., 2022), LightGCN (He et al., 2020b) learn item embeddings as (functions of) free vectors. This makes them unsuitable for making prediction for a novel test point. Other GCN-based methods such as OAK (Mohan et al., 2015), PINA (Chien et al., 2023), GraphSAGE (Hamilton et al., 2018) and GraphFormers (Yang et al., 2021) learn node representations as functions of node metadata e.g. textual descriptions. This allows the methods to work in zero-shot settings but they still incur the high storage and computational cost of GCNs. Moreover, diminishing returns are observed with increasing number of layers of the GCN (Mittal et al., 2021b; Chiang et al., 2019) with at least one model, namely LightGCN (He et al., 2020b)

foregoing all non-linearities in its network, effectively opting for a single-layer GCN. It must be noted that GCN's can be highly accurate if one can have an oracle to predict relevant nodes(table 2). However, such oracle is never available online and the slightest error in first stage retrieval leads to poor retrieval quality. (sec. 4).

We now develop the RAMEN method that offers a far more scalable alternative to GCNs and other popular graph-based architectures in XC settings, significantly reducing the overheads of graph-based learning, yet offering sustained and significant performance boosts in prediction accuracies.

## 3 RAMEN: gRAPH REGULARIZED ENCODER TRAINING FOR EXTREME CLASSIFICATION

**Notation:** Let $L$ be the total number of labels in the application. Note that the label set remains same across training and testing. Let $\mathbf{x}_i, \mathbf{z}_l$ be the textual descriptions of the data point $i$ and label $l$ respectively. For each data point $i \in [N]$, its ground truth label vector is $\mathbf{y}_i \in \{-1, +1\}^L$, where $y_{il} = +1$ if label $l$ is relevant to the data point $i$ and otherwise $y_{il} = -1$. The training set is comprised of $N$ labeled data points and $L$ labels as $\mathcal{D} := \{\{\mathbf{x}_i, \mathbf{y}_i\}_{i=1}^N, \{\mathbf{z}_l\}_{l=1}^L\}$. Let $\mathcal{X} \stackrel{\text{def}}{=} \{\mathbf{x}_i\}_{i=1}^N$ denote the set of training data points and $\mathcal{Z} \stackrel{\text{def}}{=} \{\mathbf{z}_l\}_{l=1}^L$ denote the set of labels. The meta-data graph over the auxiliary sets $\mathcal{A}$ (hyper-links, co-bidded queries) is denoted by $\mathcal{G}_{XA}$ and $\mathcal{G}_{ZA}$ for data point (document) and label respectively.

**Metadata Graphs**: RAMEN obtains metadata graphs over Anchor Sets. Let $\mathcal{A} = \{\mathbf{a}_1, \mathbf{a}_2, \ldots, \mathbf{a}_M\}$ denote an anchor set of $M$ elements *e.g.* hyperlink and category for LF-WikiSeeAlsoTitles-320K dataset. We abuse notation to let $\mathbf{a}_m$ denote the textual representation of anchor item $m \in [M]$ as well. Two distinct types of metadata graphs are possible over an anchor set:

1. Datapoint-anchor set: This is denoted as $\mathcal{G}_{XA} = (V_{XA}, E_{XA})$ with $V_{XA} \stackrel{\text{def}}{=} \mathcal{X} \cup \mathcal{A}$ i.e., the union of training data points and anchor points. The matrix $E_{XA} = \{e_{im}\} \in \{0, 1\}^{N \times M}$ encodes whether data point $\mathbf{x}_i$ has an edge to to anchor item $\mathbf{a}_m$ or not.

2. Label-anchor set: This is denoted as $\mathcal{G}_{ZA} = (V_{ZA}, E_{ZA})$ with $V_{ZA} \stackrel{\text{def}}{=} \mathcal{Z} \cup \mathcal{A}$ i.e. the union of labels and anchor points. The matrix $E_{ZA} = \{e_{lm}\} \in \{0, 1\}^{L \times M}$ encodes whether label $\mathbf{z}_l$ has an edge to anchor item $\mathbf{a}_m$ or not.

We refer the reader to Section 4 for details of how the metadata graphs are constructed using random walks. RAMEN can work with multiple anchor sets as well. For instance, given two anchor sets $\mathcal{A}^1 = \{\mathbf{a}_1^1, \mathbf{a}_2^1, \ldots, \mathbf{a}_{M_1}^1\}$ and $\mathcal{A}^2 = \{\mathbf{a}_1^2, \mathbf{a}_2^2, \ldots, \mathbf{a}_{M_2}^2\}$, a total of 4 meta data graphs are possible.

**Intuition behind RAMEN**: A popular way to incorporate graph information into XC and recommendation tasks is to take initial embeddings of a data point from an encoder and use a graph convolution step to obtain augmented embeddings for the data point. For example, let $X \in \mathbb{R}^{N \times D} = [\mathbf{x}_1, \ldots, \mathbf{x}_N]^\top$ be the initial embeddings of the $N$ data points over which a graph with adjacency matrix $A \in [0, 1]^{N \times N}$ is present. A typical layer in a GCN performs an operation of the form $\phi(AXW) \in \mathbb{R}^{N \times D}$ where $W \in \mathbb{R}^{D \times D}$ is a transformation matrix and $\phi : \mathbb{R} \to \mathbb{R}$ is some activation function applied coordinate-wise. Not only is this step expensive (Zeng et al., 2020; Hamilton et al., 2018), but also offers diminishing returns with increasing number of layers (Chiang et al., 2019; Mittal et al., 2021b). Theorem 1 indicates that in cases where the adjacency matrix can be well-predicted using a non-GCN network (say feedforward or transformer) over the initial features, the convolutional layer can be well approximated by a non-GCN network as well. Note that edge prediction is often possible with high accuracy since the metadata graph available is closely linked to the prediction task at hand and Table 6 confirms this for the tasks considered in this paper. RAMEN uses this result to infer that it may be less useful to perform graph convolutions on top of a reasonably powerful encoder such as a transformer. Instead, utilizing the graph for regularization is cheaper yet effective. Theorem 1 is specified and proved in Appendix E. Extensions of Theorem 1 to networks with multiple GCN layers are also discussed.

**Theorem 1** (Informal). *Let there exist a non-GCN (e.g. feedforward, transformer etc) network $\mathcal{F} : \mathcal{X} \to S^{P-1}$ where $S^{P-1}$ is the the unit sphere in $\mathbb{R}^P$, that effectively predicts edges in the metadata graph for any $i, j \in [N], a_{ij} \approx (1 + \mathcal{F}(\mathbf{x}_i)^\top \mathcal{F}(\mathbf{x}_j))/2$ where $A = [a_{ij}]$ is the adjacency matrix of the graph, then there exists another non-GCN network $\mathcal{H}$ such that $\phi(AXW) \approx [\mathcal{H}(\mathbf{x}_1), \ldots, \mathcal{H}(\mathbf{x}_N)]^\top$.*

**Regularization Framework**: RAMEN's training (Figure 1) consists of two main components: (a) Any XC or dense retrieval method ($\mathcal{M}$), comprising an encoder block ($\mathcal{E}_{\boldsymbol{\theta}}$), and (b) The metadata graph ($\mathcal{A}_c$). The encoder $\mathcal{E}_{\boldsymbol{\theta}} : \mathcal{X} \rightarrow \mathcal{S}^{D-1}$ with trainable parameters $\boldsymbol{\theta}$ is used to embed data points and labels using their textual descriptions. $\mathcal{S}^{D-1}$ denotes the $D$-dimensional unit sphere, i.e., the encoder provides unit-norm embeddings (unless stated otherwise). For the sake of brevity, we use $\mathcal{E}(\cdot)$ to denote the encoder. RAMEN uses a DistilBERT (Sanh et al., 2019) encoder as $\mathcal{E}$ and regularizes using the proposed training approach to learn robust and accuracy embedding representation.

**Metadata Graph Regularizers**: Given an anchor set $\mathcal{A}$ and graphs $\mathbf{G}_{XA}, \mathbf{G}_{ZA}$, we define the following two regularization functions over the encoder parameters:

$$\mathcal{R}_x(\boldsymbol{\theta}) = \sum_{i=1}^{N} \sum_{\substack{p:e_{ip}=1 \\ n:e_{in}=0}} [\mathcal{E}_{\boldsymbol{\theta}}(\mathbf{x}_i)^{\top}\mathcal{E}_{\boldsymbol{\theta}}(\mathbf{a}_n) - \mathcal{E}_{\boldsymbol{\theta}}(\mathbf{x}_i)^{\top}\mathcal{E}_{\boldsymbol{\theta}}(\mathbf{a}_p) + \gamma]_{+} \tag{1}$$

$$\mathcal{R}_z(\boldsymbol{\theta}) = \sum_{l=1}^{L} \sum_{\substack{p:e_{lp}=1 \\ n:e_{ln}=0}} [\mathcal{E}_{\boldsymbol{\theta}}(\mathbf{z}_l)^{\top}\mathcal{E}_{\boldsymbol{\theta}}(\mathbf{a}_n) - \mathcal{E}_{\boldsymbol{\theta}}(\mathbf{z}_l)^{\top}\mathcal{E}_{\boldsymbol{\theta}}(\mathbf{a}_p) + \gamma]_{+} \tag{2}$$

Here, $p$ is the positive anchor and $n$ are in-batch negatives anchors (explained in later section). Note that these two regularizers encourage the encoder to keep data points and labels closely embedded to their related anchor points and far away from unrelated anchor points. If we have more than one anchor set, say $\mathcal{A}^1, \mathcal{A}^2$, we can define corresponding regularizers $\mathcal{R}_x^t(\boldsymbol{\theta}), \mathcal{R}_z^t(\boldsymbol{\theta}), t=1, 2$.

**RAMEN Training**: RAMEN performs regularization of the encoder for any $\mathcal{M}$. The encoder is trained using document-label loss ($\mathcal{L}(\boldsymbol{\theta})$) regularized using two components: a) Anchor set on document side ($\mathcal{R}_x(\boldsymbol{\theta})$), and b) Anchor sent on label side ($\mathcal{R}_z(\boldsymbol{\theta})$), as discussed in the previous section. The $\mathcal{L}(\boldsymbol{\theta})$ function takes the following formulation:

$$\mathcal{L}(\boldsymbol{\theta}) = \sum_{i=1}^{N} \sum_{\substack{l:y_{il}=+1 \\ k:y_{ik}=-1}} [\mathcal{E}_{\boldsymbol{\theta}}(\mathbf{z}_k)^{\top}\mathcal{E}_{\boldsymbol{\theta}}(\mathbf{x}_i) - \mathcal{E}_{\boldsymbol{\theta}}(\mathbf{z}_l)^{\top}\mathcal{E}_{\boldsymbol{\theta}}(\mathbf{x}_i) + \gamma]_{+},$$

Note that this loss function encourages the encoder to embed a data point close to its relevant labels and far from irrelevant ones. The encoder is trained by minimizing the following regularized objective

$$\min_{\boldsymbol{\theta}} \left\{ \lambda_l \cdot \mathcal{L}(\boldsymbol{\theta}) + \sum_{t=1}^{T}(\lambda_x^t \cdot \mathcal{R}_x^t(\boldsymbol{\theta}) + \lambda_z^t \cdot \mathcal{R}_z^t(\boldsymbol{\theta})) \right\}$$

where $\lambda_l, \lambda_x^t, \lambda_z^t$ are regularization constants that are estimated using a bandit optimization strategy described below. This step can accommodate multiple anchor sets as well as regularizers. Once the regularized encoder training is complete for $\mathcal{M}$, the trained encoder can subsequently be used to train subsequent modules in $\mathcal{M}$, if present. For instance, XC approaches further train a per-label classifier.

**Bandit Learning for Regularization Constants:** The gradient descent without a gradient approach (Flaxman et al., 2005) was adopted to tune the regularization constants $\lambda_l, \lambda_{t,x}, \lambda_{t,z}$ in an online manner. $\lambda_l$ was initialized to 1 and $\lambda_{t,x}, \lambda_{t,z}$ to 0.1. Below we describe the process for a single constant $\lambda$ and the same is independently replicated for all the constants.

After every 30 iterations, the value of lambda is perturbed as $\hat{\lambda} = \lambda + z, z \sim \mathcal{N}(0, 0.01)$, where $\mathcal{N}(0, 0.01)$ denotes a unidimensional Gaussian with zero mean and variance 0.01. Subsequently, $\hat{\lambda}$ is used as the regularization constant in the loss expression for the next 30 iterations. The mini-batch objective values ($\lambda_l \cdot \mathcal{L} + \sum_{t=1,2}(\lambda_x^t \cdot \mathcal{R}_x^t + \lambda_z^t \cdot \mathcal{R}_z^t)$) incurred in these 30 iterations are calculated as $P$, and $\lambda$ is updated by using the estimated gradient as follows

$$\lambda = \lambda - \eta \cdot \frac{P}{\hat{\lambda} - \lambda},$$

where $\eta$ is a learning rate. This is justified by a simple but surprising application of Stokes theorem (Flaxman et al., 2005), which states that for any function $f : \mathbb{R} \rightarrow \mathbb{R}$ (which can itself be non-convex or even non-differentiable), we have $\frac{d\hat{f}(x)}{dx} = \frac{1}{\delta} \cdot \mathbb{E}_{u \sim \{-1, +1\}} [f(x + \delta u)u]$ where $\hat{f} : \mathbb{R} \rightarrow \mathbb{R}$ is a smoothed version of $f$ defined as $\hat{f}(x) \stackrel{\text{def}}{=} \mathbb{E}_{v \sim [-1, 1]} [f(x + \delta v)]$. Note $\hat{f}$ is always differentiable even if $f$ is not. In order to compute the mini-batch objectives, $P$, RAMEN mines hard negatives. The negative mining technique is explained below.

**Negative Mining:** The loss function and regularizers contain $\mathcal{O}\left(NL \log L + (N + L)M \log M\right)$ terms where $M = \max \{M_1, M_2\}$ is the maximum number of anchors in any of the anchor sets. This is because the number of relevant labels per data point is usually limited by $|l : y_{il} = +1| \leq \mathcal{O}\left(\log L\right)$ in XC applications (Jain et al., 2016) and we can construct the metadata graphs to have at most $\log M$ relevant anchors per data point or label. Performing optimization with respect to all these terms is expensive which is why RAMEN utilizes in-batch negative mining (Guo et al., 2019; Dahiya et al., 2021a; 2023; Faghri et al., 2018; Chen et al., 2020; He et al., 2020a). Specifically, a set of data points is identified and for each data point, and a random relevant label and random related anchor are chosen (from each anchor set if there are multiple anchor sets). For each of the chosen labels, a random related anchor is chosen from each anchor set. Then, hard negative labels for a data point are chosen only amongst those labels selected for that particular mini-batch. Similarly, hard negative anchors for a data point or label are chosen from only those anchors selected for that mini-batch.

**Inference with RAMEN:** RAMEN's training framework is applied to the encoder in the base model ($\mathcal{M}$). Once training is complete, inference remains unchanged from the base model's proposed approach. Here, RAMEN incurs no additional inference time over the base method and improves accuracy by 2-3% in P@1.

## 4 EXPERIMENTS

The XML Repository (Bhatia et al., 2016) provides various public XC datasets which are thoroughly studied and benchmarked by plethora of papers. These datasets are curated from Wiki dumps link and Amazon dump (Ni et al., 2019) but graph metadata, which was readily available, was ignored. For RAMEN, we crawl these dumps and curate metadata for all public datasets as follows:

**LF-WikiSeeAlsoTitles-320K**: The dataset was curated from Wiki dump **link**. The scenario involved recommending related articles. Articles under the "See Also" section were used as ground truth labels. Internal hyperlinks and category links were used to create two sets of metadata graphs, one using hyperlinked Wikipedia articles as anchors and the other using Wikipedia categories as anchors.

**LF-WikiTitles-500K**: The dataset was also curated from Wiki dump **link**. The scenario involved recommending relevant "categories" for an article. Internal hyperlinks and category-to-category links were used to create two sets of metadata graphs as described above.

**LF-AmazonTitles-1.3M**: The dataset was curated from Amazon dump (Ni et al., 2019). The scenario involved recommending relevant "products" for a product. The "similar_items" links given in the data dump were used to create the metadata graph.

**Dataset**: Please refer to Tab. 15 of the appendix for dataset statistics. For all datasets, test data points were removed from the graph if present as nodes to prevent train-test leaks.

**Implementation details**: We initialize the encoder with a pre-trained DistilBERT and fine-tune it. The metadata graphs are pruned using the fine-tuned encoder. Table 16 in the appendix summarizes all hyper-parameters for each dataset. It is notable that even though RAMEN uses a graph at training time, inference does not require any such information, making it highly suitable for long-tail queries. We compare three variants of RAMEN against baseline XC and dense retrieval approaches. In particular, we consider RAMEN (ANCE), RAMEN (NGAME) and RAMEN (DEXML). All RAMEN variants and most baseline variants use the PyTorch (Paszke et al., 2017) framework and were trained on 4 Nvidia V100 GPUs. DEXML (Gupta et al., 2023) was trained on 16 Nvidia A100 GPUs. Refer to Appendix B for additional details.

**Results on benchmark datasets**: Table 1 compares RAMEN variants with graph and XC methods. RAMEN is 5% more accurate over the best baseline numbers. In particular RAMEN is 2-3% more accurate than traditional graph-based methods. Additionally, the RAMEN variants are 3-4% more accurate over OAK (Mohan et al., 2015) & PINA (Chien et al., 2023), which use both XC and graph metadata. RAMEN's primary focus is short-text documents but for results on full text datasets refer to Table 14 in appendix.

**Analysis of gains**: *Note that, Theorem 1 states that RAMEN and GCNs are equivalent.* However, as discussed in **Limitation of GCN Methods** in the introduction, GCN's two stage retrieval pipeline can be noisy. Table 2 demonstrates that if we replace the first stage with the oracle linker (first statge with zero error), the performance of these graph-based methods starts to outperform RAMEN variants.

Table 1: Results on short-text benchmark datasets. RAMEN variants is up to 15% more accurate as compared to both text-based and graph-based baselines. For details on evaluation metrics, please refer to section D in appendix.

| | PSP@1 | PSP@3 | PSP@5 | PSN@3 | PSN@5 | P@1 | P@3 | P@5 | N@3 | N@5 |
|---|---|---|---|---|---|---|---|---|---|---|
| | | | | | LF-AmazonTitles-1.3M | | | | | |
| RAMEN (ANCE) | **37.0** | **40.0** | **41.2** | **39.3** | **40.5** | 48.7 | 42.9 | 38.4 | 47.3 | 46.3 |
| RAMEN (NGAME) | 34.3 | 37.4 | 39.0 | 36.9 | 38.4 | 55.6 | 49.7 | 44.9 | 54.3 | 53.3 |
| RAMEN (DEXML) | 31.2 | 34.7 | 36.7 | 34.2 | 36.1 | **58.8** | **51.1** | **45.8** | **56.1** | **54.6** |
| GraphSage | 24.5 | 24.2 | 23.7 | 24.7 | 24.9 | 28.1 | 21.4 | 17.6 | 24.8 | 23.2 |
| GraphFormer | 22.5 | 22.4 | 22.5 | 22.6 | 23.1 | 24.2 | 17.4 | 14.3 | 21.6 | 20.8 |
| DEXML | - | - | 36.6 | - | - | 58.6 | 50.9 | 45.6 | 55.9 | 54.4 |
| NGAME | 29.2 | 33.0 | 35.4 | 32.1 | 33.9 | 56.7 | 49.2 | 44.1 | 53.8 | 52.4 |
| DEXA | 29.1 | 32.7 | 34.9 | 32.0 | 33.9 | 56.6 | 49.0 | 43.9 | 53.8 | 52.4 |
| ANCE | 33.1 | 35.6 | 36.8 | - | - | 45.8 | 39.9 | 35.5 | - | - |
| CascadeXML | 17.2 | 21.7 | 24.8 | 19.9 | 21.5 | 47.8 | 42.0 | 38.3 | 45.0 | 43.8 |
| XR-Transformer | 20.1 | 24.8 | 27.8 | 23.4 | 25.4 | 50.1 | 44.1 | 40.0 | 47.7 | 46.6 |
| PINA | - | - | - | - | - | 55.8 | 48.7 | 43.9 | - | - |
| AttentionXML | 16.0 | 19.9 | 22.5 | 18.2 | 19.6 | 45.0 | 39.7 | 36.2 | 42.4 | 41.2 |
| SiameseXML | 27.1 | 30.4 | 32.5 | 29.4 | 30.9 | 49.0 | 42.7 | 38.5 | 46.4 | 45.1 |
| ECLARE | 23.4 | 27.9 | 30.6 | 26.7 | 28.6 | 50.1 | 44.1 | 40.0 | 47.7 | 46.7 |
| | | | | | LF-WikiTitles-500K | | | | | |
| RAMEN (ANCE) | **30.5** | 26.9 | 25.7 | 30.0 | 31.4 | 46.1 | 25.4 | 17.4 | 35.4 | 33.8 |
| RAMEN (NGAME) | 30.1 | **27.4** | **26.4** | **30.2** | **31.7** | **48.2** | **27.4** | **19.0** | **37.6** | **35.9** |
| OAK | 25.7 | 25.8 | 25.0 | 27.8 | 29.4 | 44.8 | 25.9 | 17.9 | 35.4 | 33.8 |
| GraphSage | 22.3 | 19.3 | 19.1 | 22.1 | 23.8 | 27.2 | 15.7 | 11.3 | 22.6 | 22.8 |
| GraphFormer | 22.0 | 19.2 | 19.5 | 21.3 | 22.8 | 24.5 | 14.9 | 11.3 | 20.2 | 20.3 |
| NGAME | 23.1 | 23.3 | 23.0 | 25.3 | 27.2 | 39.0 | 23.1 | 16.1 | 31.8 | 30.7 |
| ANCE | 23.2 | 22.1 | 21.2 | 24.5 | 26.1 | 29.7 | 18.1 | 12.5 | 25.4 | 25.1 |
| CascadeXML | 19.2 | 19.5 | 19.7 | 20.8 | 22.3 | 47.3 | 26.8 | 19.0 | 36.2 | 34.4 |
| AttentionXML | 14.8 | 14.0 | 13.9 | 15.2 | 16.2 | 40.9 | 21.5 | 15.0 | 29.4 | 27.4 |
| ECLARE | 21.6 | 20.4 | 19.8 | 22.4 | 23.6 | 44.4 | 24.3 | 16.9 | 33.3 | 31.5 |
| | | | | | LF-WikiSeeAlsoTitles-320K | | | | | |
| RAMEN (ANCE) | **29.0** | **31.8** | **34.5** | **31.7** | **33.6** | 35.2 | 24.0 | 18.4 | 35.2 | 36.5 |
| RAMEN (NGAME) | 28.6 | 31.6 | 34.4 | 31.5 | 33.5 | **35.5** | **24.3** | **18.6** | **35.6** | **36.8** |
| OAK | 25.8 | 28.5 | 30.8 | 28.6 | 30.3 | 33.7 | 22.7 | 17.1 | 33.4 | 34.4 |
| GraphSage | 21.6 | 21.8 | 23.5 | 22.9 | 24.6 | 27.3 | 17.2 | 13.0 | 27.1 | 28.4 |
| GraphFormer | 19.2 | 20.6 | 22.7 | 21.0 | 22.7 | 21.9 | 15.1 | 11.8 | 22.6 | 24.0 |
| NGAME | 24.4 | 27.4 | 29.9 | 27.4 | 29.2 | 32.6 | 22.0 | 16.6 | 32.3 | 33.2 |
| DEXA | 24.4 | 26.5 | 28.6 | 27.0 | 28.6 | 31.7 | 21.0 | 15.8 | 31.3 | 32.3 |
| DEXML | 22.8 | 23.9 | 25.7 | 25.1 | 26.7 | 29.9 | 19.7 | 14.8 | 29.7 | 30.7 |
| ANCE | 25.1 | 26.8 | 28.7 | 27.3 | 28.9 | 30.8 | 20.3 | 15.4 | 30.5 | 31.5 |
| ELIAS | 13.5 | 15.9 | 17.7 | 15.6 | 16.8 | 23.4 | 15.6 | 11.8 | 24.7 | 23.6 |
| CascadeXML | 12.7 | 15.4 | 17.6 | 14.6 | 16.0 | 23.4 | 15.7 | 12.1 | 22.6 | 23.4 |
| XR-Transformer | 10.6 | 11.8 | 12.7 | 11.7 | 12.4 | 19.4 | 12.2 | 9.0 | 18.3 | 18.5 |
| AttentionXML | 9.4 | 10.6 | 11.7 | 10.4 | 11.2 | 17.6 | 11.3 | 8.5 | 16.6 | 17.1 |
| SiameseXML | 26.8 | 28.4 | 30.4 | 28.7 | 30.3 | 32.0 | 21.4 | 16.2 | 31.6 | 32.6 |
| ECLARE | 22.0 | 24.2 | 26.3 | 24.5 | 26.0 | 29.3 | 19.8 | 15.0 | 29.2 | 30.2 |

Table 2: Results using Oracle Linker for GCN Vs RAMEN (ANCE) on LF-WikiSeeAlsoTitles-320K.

| Method | P@1 | P@5 | N@5 | PSP@1 | PSP@5 |
|---|---|---|---|---|---|
| RAMEN (ANCE) | 35.2 | 18.4 | 35.2 | 29.0 | 34.5 |
| OAK | 33.7 | 17.1 | 34.4 | 25.8 | 30.8 |
| OAK + Oracle | 38.9 | 19.4 | 40.4 | 29.7 | 34.8 |

Table 3: RAMEN (ANCE)'s computational relative to baselines on LF-WikiSeeAlsoTitles-320K.

| Method | Train Time | Pred Time | Model size | P@1 |
|---|---|---|---|---|
| RAMEN (ANCE) | 1× | 1× | 1× | 35.2 |
| OAK | 1.5× | 2× | 3.5× | 33.7 |
| ANCE | 0.9× | 1× | 1× | 30.8 |
| DEXML | 2.1× | 1× | 1× | 29.9 |

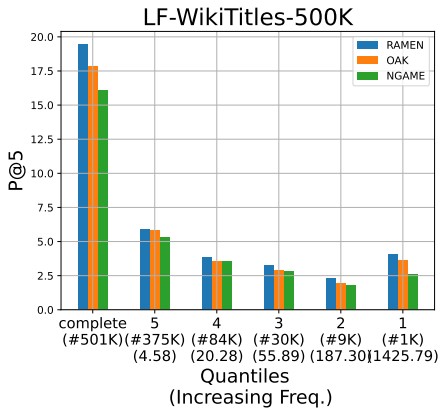

Table 4: Quantile wise-comparison of RAMEN and other methods. RAMEN (NGAME) gives consistent gains in each bin (see Appendix D for binning details). The left-most bin contains the most rare/tail labels whereas the rightmost bin contains the most popular/head labels.

Table 5: Ablations were done using ANCE as base algorithm (RAMEN (ANCE)) on LF-WikiSeeAlsoTitles-320K to understand the impact of design choices on the quality of encoder training. RAMEN (ANCE)'s design choices are seen to be optimal and offer 2.5–13% improvement in the P@1 metric over alternate design choices.

| RAMEN | P@1 | P@3 | P@5 | N@3 | N@5 |
|---|---|---|---|---|---|
| RAMEN (ANCE) | 35.2 | 24.1 | 18.3 | 35.3 | 36.5 |
| − No Bandits | 20.9 | 12.8 | 9.5 | 21.5 | 22.5 |
| − No Pruning | 31.3 | 18.9 | 12.8 | 31.43 | 31.5 |
| − No Doc. Graph | 29.7 | 17.5 | 12.5 | 30.7 | 30.8 |
| − No Lbl. Graph | 34.1 | 22.7 | 14.6 | 32.0 | 34.1 |
| AugGT | 15.6 | 8.9 | 6.5 | 15.7 | 16.3 |
| graph-init=0.1 | 35.2 | 24.1 | 18.4 | 35.3 | 36.5 |
| graph-init=0.5 | 34.5 | 24.4 | 19.3 | 35.2 | 36.4 |
| graph-init=1.0 | 34.8 | 24.8 | 17.7 | 36.2 | 35.6 |

Table 6: RAMEN (ANCE)'s encoder $\mathcal{E}$ can predict links in the meta-data graph with high recall (R@100).

| Link type | LF-WikiSeeAlsoTitles-320K | LF-WikiTitles-500K |
|---|---|---|
| hyper_link | 99.88 | 94.20 |
| category | 99.88 | 99.96 |

Table 7: RAMEN (ANCE)'s performance on P and PSP decreases as the volume of metadata decreases for LF-WikiSeeAlsoTitles-320K

| RAMEN (ANCE) | PSP@1 | PSP@5 | P@1 | P@5 |
|---|---|---|---|---|
| 100% | 29.0 | 34.5 | 35.2 | 18.4 |
| 50% | 26.4 | 31.1 | 33.8 | 17.7 |
| 20% | 25.8 | 30.1 | 33.1 | 17.1 |

Table 8: Comparing the difference in false negative rates between RAMEN (ANCE) and OAK on LF-WikiSeeAlsoTitles-320K. Results are presented quantile wise. The #5K label quantile contains the most popular/head labels whereas the #729K quantile contains the most rare/tail labels. See Appendix D for definitions of metrics and quantiles. RAMEN (ANCE)'s false negative rate is always superior to that of OAK but the gap widens significantly when compared on tail labels.

| Quantile (#L) Avg. Doc. | DIFF@5 | DIFF@10 | DIFF@20 | DIFF@50 | DIFF@100 |
|---|---|---|---|---|---|
| (#1K) 200.78 | -0.019 | -0.026 | -0.027 | -0.021 | -0.010 |
| (#9K) 30.98 | 0.013 | -0.050 | -0.107 | -0.123 | -0.049 |
| (#31K) 9.14 | 0.036 | -0.124 | -0.272 | -0.443 | -0.423 |
| (#74K) 3.94 | -0.455 | -0.757 | -1.091 | -1.551 | -1.854 |
| (#195K) 1.49 | -2.246 | -2.722 | -3.529 | -4.763 | -6.075 |
| complete | -2.671 | -3.679 | -5.025 | -6.901 | -8.411 |

Table 9: Results of RAMEN as a regularizer in baseline algorithms on LF-WikiSeeAlsoTitles-320K. RAMEN's regularization algorithm improves the respective performance by 2–3%.

| | P@1 | P@3 | P@5 | N@3 | N@5 |
|---|---|---|---|---|---|
| ANCE | 30.8 | 20.3 | 15.4 | 30.5 | 31.5 |
| RAMEN (ANCE) | 35.2 | 24.1 | 18.3 | 35.3 | 36.5 |
| NGAME | 32.6 | 22.0 | 16.6 | 32.3 | 33.2 |
| RAMEN (NGAME) | 35.5 | 24.3 | 18.6 | 35.6 | 36.8 |
| SiameseXML | 31.9 | 21.4 | 16.2 | 31.6 | 32.6 |
| RAMEN (SiameseXML) | 32.0 | 21.9 | 17.6 | 31.7 | 32.9 |
| ECLARE | 29.4 | 19.8 | 15.1 | 29.2 | 30.2 |
| RAMEN (ECLARE) | 30.5 | 20.1 | 16.4 | 32.3 | 32.8 |

Table 10: Impact of different graph metata on RAMEN (ANCE)'s performance

| Method | P@1 | P@5 | N@5 | PSP@1 | PSP@5 | PSN@5 |
|---|---|---|---|---|---|---|
| RAMEN (ANCE) | 35.2 | 18.4 | 36.5 | 29.0 | 34.5 | 33.6 |
| Only category | 33.7 | 17.4 | 34.4 | 27.6 | 32.7 | 31.7 |
| Only hyperlink | 34.2 | 17.6 | 35.0 | 27.8 | 32.7 | 32.0 |

Table 11: Results on the proprietary dataset (G-EPM-1M). RAMEN (Method-1) is ≈10% more accurate than the leading method in production.

| | PSP@1 | PSP@3 | P@1 | P@5 | R@10 |
|---|---|---|---|---|---|
| RAMEN (Method-1) | **25.1** | **47.2** | **25.2** | **9.8** | **55.9** |
| Method-1 | 16.2 | 31.5 | 15.1 | 6.3 | 37.2 |

Table 12: RAMEN (ANCE)'s performance in zero shot scenario

| Method | P@1 | P@3 | P@5 | N@3 | N@5 |
|---|---|---|---|---|---|
| LF-WikiSeeAlsoTitles-320K | | | | | |
| RAMEN (ANCE) | **14.8** | **10.2** | **7.3** | **19.8** | **21.4** |
| ANCE | 12.1 | 8.5 | 6.06 | 16.3 | 17.6 |
| LF-WikiTitles-500K | | | | | |
| RAMEN (ANCE) | **11.4** | **5.3** | **3.5** | **12.1** | **12.5** |
| ANCE | 10.9 | 5.1 | 3.5 | 11.4 | 11.9 |

Table 13: A subjective comparison of predictions made by RAMEN, the leading text-based method NGAME, and the leading graph-based method GraphFormers on LF-WikiSeeAlsoTitles-320K. Labels that are a part of the ground truth are formatted in black color. Labels not a part of the ground truth are formatted in light gray color. Relevant labels that are missing from the ground truth are marked in bold black. RAMEN (ANCE) could make predict highly relevant labels such as "Crown group", which were missing from the ground truth as well as omitted by other methods.

| Method | Prediction |
|---|---|
| Document: Clade | |
| RAMEN (ANCE) | Cladistics, Phylogenetics, **Crown group**, Paraphyly, Polyphyly |
| ANCE | Cladistics, Linnaean taxonomy, **Polyphyly**, Paragroup, Molecular phylogenetics |
| OAK | Phylogenetic nomenclature, Molecular phylogenetics, Haplotype, Cladistics, Paragroup |

However, this oracle linker is never available for a novel test point, and RAMEN variants achieved a similar performance in a fraction of the cost of training and prediction time as shown in Table 3. Additionally RAMEN variants can predict meta-data graph links with high accuracy (Table 6) which validates the proposed Theorem 1. Furthermore, Figure 4 shows that RAMEN variants outperform baseline methods in each quantile, showing its overall superior embedding quality. To demonstrate the zero-shot performance of RAMEN, we consider RAMEN (ANCE), since it is a dense retrieval approach. RAMEN (ANCE) achieves state of the art performance in the zero-shot scenario (cf. Table 12) indicating that RAMEN (ANCE)'s embeddings are robust to unseen labels. While RAMEN variants outperform baseline methods on all datasets, the low gains in LF-AmazonTitles-1.3M can be attributed to the low volume of metadata ("similar_items" graph edges) available for training (cd. Table 15 in the appendix). To validate this, experiments on the LF-WikiSeeAlsoTitles-320K dataset were conducted where it was observed that reducing metadata to 50% and 20% resulted in performance drops of 1–2.5% in PSP and 3–4% in P, respectively. This emphasizes the importance of metadata for performance (Table 7).

Table 13 shows that RAMEN (ANCE) could make predictions such as "Crown group" which were missing labels in the training data, by exploiting metadata graph links. Apart from standard metrics, the error rate plays a crucial role in deployment. Tab. 8 compares the difference in RAMEN (ANCE)'s false negative rate (FN@$k$) with the best-performing baseline (OAK) when each method was allowed to make $k$ predictions i.e., Diff@$k \stackrel{\text{def}}{=}$ FN@$k$(RAMEN) - FN@$k$(OAK). It is notable that RAMEN (ANCE) consistently outperforms OAK over all label quantiles (i.e. over head/popular as well as tail/rare labels) and performs even better at higher values of $k$ such as $k = 50$.

**Case-study for Sponsored Search**: Matching user queries with relevant advertiser keywords is a critical component of sponsored search. One type of matching is Extended Phrase Match (EPM), which aims to match a user query with advertiser keywords that have a subset of the query's intent. This means that only keywords with similar intent to the query are considered. For example, for the query "cheap nike shoes", a valid EPM keyword is "nike sneakers" but "adidas shoes" or "nike shorts" are not.

We study the effectiveness of RAMEN (Method-1) in this application by comparing it against the state-of-the-art encoder in production (anonymised as Method-1) and also conducting A/B test on live

search-engine traffic. For offline comparison, the G-EPM-1M dataset was curated by analyzing the ad click logs. Further, the click logs were mined to gather graph metadata for RAMEN (Method-1), including two types of signals:

1. Co-session queries: Queries that were asked in the same search session by multiple users.
2. Co-clicked queries: Queries that resulted in clicks on the same webpage.

RAMEN (Method-1) was found to be 3% better on the P@5 than Method-1. RAMEN (Method-1) was further found to be 15% better than Method-1 on the propensity scored PSP@5, indicating that RAMEN (Method-1) could match tail keywords more accurately. Please refer to Table 11. The quality of the two models was further measured using an in-production, large cross-encoder oracle quality model that was trained on an extensive set of manually labeled data. The oracle quality model found predictions made by RAMEN (Method-1) to be 43% more accurate than those by Method-1.

RAMEN (Method-1) was trained on a dataset containing 540M training documents and 360M labels mined as described above but over a longer period to conduct an A/B test on the search engine. RAMEN (Method-1) was found to increase the Impression-Yield(relevant ad impressions per user query) by 2.8% and the Click-Yield(clicks per user query) by 2.5% when compared against a control containing state-of-the-art embedding-based, generative, GCN, and XC algorithms.

**Ablations on the Design**: Experiments were conducted to understand the impact of the design choices made by RAMEN (ANCE) as well as the impact of metadata on the performance of RAMEN (ANCE). Table 5 complies these experiments. In particular, experiment "No-Bandits" explored the effects of different choices of giving weights to different sources of metadata on RAMEN's performance. A key finding was that when uniform weights were assigned to all graphs, there was a substantial 18% drop in P@1. In addition to that, similarly accuracy of RAMEN (ANCE) for different initialization of graph weights proved robustness of bandit learning. This highlights the importance of bandit learning, where each graph's contribution is determined dynamically. RAMEN (ANCE)'s robust training strategy extends benefits beyond its own performance. NGAME, SiameseXML and ECLARE, baseline methods, experienced improvements of 3% and 5% in P@1, respectively, when leveraging RAMEN (ANCE)'s metadata regularizer. (Table 9).

**Ablations on the Graph**: To understand the impact of noisy edges in metadata, experiment "No Pruning" disabled the trimming of noisy edges using cosine similarity filtering. A 4% loss in P@1 was observed which underscores the necessity of pruning unhelpful edges during training. RAMEN (ANCE) uses multiple meta-data graphs for both document and label. To ascertain the contributions of the anchor-doc and anchor-label metadata graphs, the ablations "No Doc. Graph" and "No Lbl. Graph" were conducted. These experiments reveal that information from these graphs plays a significant role, as disabling either leads to a 1.5–2% reduction in P@1. The information from these graphs can be incorporated in baseline methods like ANCE. To understand its impact, experiment "AugGT" trains ANCE with augmented ground truth. The ground truth was expanded by using label propagation wherein a label and a training point are linked by an edge if the label shares a neighbor in the metadata graph of the said training point. RAMEN (ANCE) outperformed the "AugGT" setup by 15%. This suggests that while leveraging graph information for ground truth enhancement is convenient, it may not be as effective due to noisy edges. RAMEN (ANCE) uses multiple sources of graph metadata, Table 10 in appendix shows the RAMEN (ANCE) benefits when all graph metadata is used but remains state of the art even if it uses only hyperlink metadata, similar to OAK. For more experimental details refer to Appendix A.

## 5 CONCLUSION

This paper presented RAMEN, a novel approach for leveraging metadata to enhance the accuracy of recommendation systems w.r.t. tail labels. A key takeaway from the study is that opting for graph-based regularization instead of the more prevalent GCN architectures, can yield gains of up to 15% in PSP@1/P@1 and up to 6% when compared with XC techniques tailored for recommendation systems. The bandit-style regularization technique adopted by RAMEN was found to offer performance boosts to baseline methods. Notably, RAMEN offers state-of-the-art performance without incurring any computational overhead during inference.

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

# Graph Regularized Encoder Training for
# Extreme Classification
## (Appendix)

## A ADDITIONAL RESULTS

Table 14: Results on full-text benchmark datasets. RAMEN is up to 15% more accurate as compared to both text-based and graph-based baselines.

| | PSP@1 | PSP@5 | PSN@5 | P@1 | P@5 | N@5 |
|---|---|---|---|---|---|---|
| LF-WikiSeeAlso-320K | | | | | | |
| RAMEN (ANCE) | **37.9** | **45.6** | **45.5** | **50.5** | 25.2 | 52.4 |
| RAMEN (NGAME) | 36.9 | 45.5 | 45.2 | 50.4 | **25.3** | **52.4** |
| OAK | 33.9 | 40.4 | 40.3 | 48.6 | 23.3 | 49.2 |
| GraphSage | 20.6 | 23.1 | 26.6 | 24.1 | 9.1 | 25.3 |
| GraphFormer | 16.9 | 20.9 | 20.4 | 18.1 | 8.8 | 20.8 |
| NGAME | 33.8 | 41.0 | 41.0 | 47.7 | 23.7 | 48.99 |
| DEXA | 31.8 | 38.9 | - | 47.1 | 22.7 | 47.6 |
| ANCE | 29.6 | 32.8 | 34.2 | 45.7 | 17.32 | 45.4 |
| CascadeXML | 22.3 | 31.1 | 28.9 | 40.4 | 20.2 | 40.6 |
| XR-Transformer | 25.2 | 33.8 | 32.6 | 42.6 | 21.3 | 43.4 |
| PINA | - | - | - | 44.5 | 22.9 | - |
| AttentionXML | 22.7 | 29.8 | 28.4 | 40.5 | 19.9 | 40.3 |
| LightXML | 17.9 | 24.2 | 22.8 | 34.5 | 16.8 | 34.2 |
| SiameseXML | 29.0 | 36.0 | 35.2 | 42.2 | 21.39 | 43.4 |
| ECLARE | 26.1 | 33.1 | 32.3 | 40.6 | 20.2 | 41.2 |
| DECAF | 25.7 | 34.9 | 33.7 | 41.4 | 21.4 | 43.3 |
| Parabel | 17.1 | 23.5 | 21.9 | 33.5 | 16.6 | 33.3 |
| Bonsai | 18.2 | 25.7 | 23.8 | 34.9 | 17.7 | 35.3 |
| LF-Wikipedia-500K | | | | | | |
| RAMEN (ANCE) | **50.9** | **61.9** | **61.8** | 81.1 | 50.1 | 75.3 |
| RAMEN (NGAME) | 43.6 | 61.8 | 60.2 | **85.9** | **52.6** | **79.2** |
| OAK | 45.3 | 60.8 | 59.9 | 85.2 | 50.8 | 77.3 |
| GraphSage | 35.2 | 37.8 | 40.8 | 43.1 | 28.3 | 35.3 |
| GraphFormer | 25.2 | 21.8 | 24.8 | 31.1 | 14 | 24.87 |
| LEVER | 42.5 | 60.2 | - | 85.1 | 52.1 | - |
| DEXML | - | 58.9 | - | 85.8 | 50.5 | 77.1 |
| NGAME | 41.3 | 57.1 | 56.1 | 84.1 | 49.9 | 75.9 |
| DEXA | 42.6 | 58.3 | 57.4 | 84.9 | 50.5 | 76.8 |
| ANCE | 50.9 | 57.3 | - | 77.9 | 40.9 | - |
| ELIAS | 35.1 | 51.1 | - | 81.3 | 48.8 | 73.1 |
| CascadeXML | 31.9 | 44.9 | 43.9 | 80.7 | 46.3 | 70.5 |
| XR-Transformer | 33.6 | 47.8 | 46.6 | 81.6 | 47.9 | 72.4 |
| PINA | - | - | - | 82.8 | 50.1 | - |
| AttentionXML | 34 | 50.2 | 47.7 | 82.7 | 50.4 | 74.7 |
| LightXML | 31.9 | 46.5 | 45.2 | 81.6 | 47.6 | 72.2 |
| SiameseXML | 33.9 | 37.1 | 38.9 | 67.3 | 33.7 | 54.3 |
| ECLARE | 31.1 | 38.3 | 34.5 | 68.1 | 35.7 | 56.4 |
| Parabel | 26.9 | 35.3 | 34.6 | 68.7 | 38.6 | 58.6 |
| Bonsai | - | - | - | 69.2 | 38.8 | - |

**Results on benchmark datasets**: Table 1 compares RAMEN with graph and XC methods. RAMEN is 5% more accurate over the best baseline numbers. In particular RAMEN is 2-3% more accurate than traditional graph-based methods. Additionally, RAMEN is 3-4% more accurate over OAK (Mohan

et al., 2015) & PINA (Chien et al., 2023), which uses both XC and graph metadata. Table 11 compares RAMEN against than best production method (Method-1) on the G-EPM-1M dataset. RAMEN was found to be 3% better on the P@5. RAMEN was further found to be 15% better than Method-1 on the propensity scored PSP@5, indicating that RAMEN could match tail keywords more accurately. The quality of the two models was further measured using an in-production, large cross-encoder oracle quality model that was trained on an extensive set of manually labeled data. The oracle quality model found predictions made by RAMEN to be 43% more accurate than those made by Method-1. Note that, RAMEN's primary focus is short-text documents but for results on full text counterparts of the dataset refer to Table 14 in the appendix.

**Analysis of gains**: *Note that, Theorem 1 states that RAMEN and GCNs are equivalent*. However, as discussed in **limitation of GCN** in the introduction, GCN's two stage retrieval pipeline can be noisy. Table 2 demonstrates that if we replace the first stage with the oracle linker (first statge with zero error), the performance of these graph-based methods starts to outperform RAMEN. However, this oracle linker is never available for a novel test point, and RAMEN achieved a similar performance in a fraction of the cost of training and prediction time as shown in Table 3. Additionally RAMEN can predict meta-data graph links with high accuracy (Table 6) which validates the proposed Theorem 1.

**Ablations on the Design**: Experiments were conducted to understand the impact of the design choices made by RAMEN as well as the impact of metadata on the performance of RAMEN. Table 5 complies these experiments. In particular, experiment "No-Bandits" explored the effects of different choices of giving weights to different sources of metadata on RAMEN's performance. A key finding was that when uniform weights were assigned to all graphs, there was a substantial 18% drop in P@1. In addition to that, similary accuracy of RAMEN for different initialization of graph weights proved robustness of bandit learning. This highlights the importance of bandit learning, where each graph's contribution is determined dynamically. RAMEN's robust training strategy extends benefits beyond its own performance. SiameseXML and ECLARE, baseline methods, experienced improvements of 3% and 5% in P@1, respectively, when leveraging RAMEN's metadata regularizer (Table 9).

**Ablations on the Graph**: To understand the impact of noisy edges in metadata, experiment "No Pruning" disabled the trimming of noisy edges using cosine similarity filtering. A 4% loss in P@1 was observed which underscores the necessity of pruning unhelpful edges during training. RAMEN uses multiple meta-data graphs for both document and label. To ascertain the contributions of the anchor-doc and anchor-label metadata graphs, the ablations "No Doc. Graph" and "No Lbl. Graph" were conducted. These experiments reveal that information from these graphs plays a significant role, as disabling either leads to a 1.5–2% reduction in P@1. The information from these graphs can be incorporated in baseline methods like ANCE. To understand its impact, experiment "AugGT" trains ANCE with augmented ground truth. The ground truth was expanded by using label propagation wherein a label and a training point are linked by an edge if the label shares a neighbor in the metadata graph of the said training point. RAMEN outperformed the "AugGT" setup by 15%. This suggests that while leveraging graph information for ground truth enhancement is convenient, it may not be as effective due to noisy edges. RAMEN uses multiple sources of graph metadata, table 10 in appendix shows the alg benefits when all graph metadata is used but remains state of the art even if it uses only hyperlink metadata, similar to OAK.

These experiments collectively shed light on the intricate interplay between design choices and metadata utilization, underscoring the effectiveness and nuances of RAMEN's approach.

# B IMPLEMENTATION DETAILS

Links obtained on the metadata graph from raw data suffer from missing links in much the same way there are missing labels in the ground truth. To deal with this, RAMEN performs a random walk with restart on each anchor node. The random walk was performed for 400 hops with a restart probability of 0.8, thus ensuring that the walk did not wander too far from the starting node. This random walk could also introduce noisy edges, leading to poor model performance. To deal with such edges, in-batch pruning was performed and edges to only those anchors were retained which had a cosine similarity of $> 0$ based on the embeddings given the encoder. To get the encoder, RAMEN initialize the encoder with a pre-trained DistilBERT and fine-tuned it for 10 epochs(warmup phase) using unpruned metadata graphs. Then the metadata graphs were pruned using the fine-tuned encoder. Encoder fine-tuning was then was continued for 5 epochs using the pruned graphs after which the graphs were re-pruned. These alternations of 5 epochs of encoder fine-tuning followed by re-pruning were repeated till convergence. The learning rate for each bandit was set to $0.01$. Table 16 in supplementary material summarizes all hyper-parameters for each dataset. It is notable that even though RAMEN uses a graph at training time, inference does not require any such information, making it highly suitable for long-tail queries.

## C DATA STATS

Table 15: Dataset statistics summary for benchmark datasets used by RAMEN. Entries marked with ‡ were not disclosed because the dataset is proprietary.

| # Train Pts $N$ | # Labels $L$ | # Test Pts $N'$ | Avg. docs. per label | Avg. labels per doc. | Graph Types | # Graph Nodes $G$ | Avg. node neighbors per doc. | Avg. node neighbors per label |
|---|---|---|---|---|---|---|---|---|
| \multicolumn{9}{c}{**LF-WikiSeeAlsoTitles-320K / LF-WikiSeeAlso-320K**} |
| 693,082 | 312,330 | 177,515 | 4.67 | 2.11 | Hyperlink | 2,458,399 | 38.87 | 7.71 |
| | | | | | Category | 656,086 | 4.74 | 4.82 |
| \multicolumn{9}{c}{**LF-WikiTitles-500K / LF-Wikipedia-500K**} |
| 1,813,391 | 501,070 | 783,743 | 17.15 | 4.74 | Hyperlink | 2,148,579 | 16.46 | 8.53 |
| | | | | | Category | 766,929 | 2.35 | 4.21 |
| \multicolumn{9}{c}{**LF-AmazonTitles-1.3M**} |
| 2,248,619 | 1,305,265 | 970,237 | 38.24 | 22.20 | related_items | 916269 | 1.98 | 3.95 |
| | | | | | category | 17981 | 3.35 | 583.04 |
| \multicolumn{9}{c}{**G-EPM-1M**} |
| 10,746,967 | 999,987 | 4,607,267 | ‡ | ‡ | Co-session queries | ‡ | ‡ | ‡ |
| | | | | | Co-click queries | | | |

Table 16: Hyper-parameter values for RAMEN on all datasets to enable reproducibility. RAMEN code will be released publicly. Most hyperparameters were set to their default values across all datasets. LR is learning rate. Multiple clusters were chosen to form a batch hence $B > C$. Clusters were refreshed after 5 epochs. Cluster size $C$ was doubled after every 25 epochs. Margin $\gamma = 0.3$ was used for contrastive loss. For training M2 number of positive samples and negative samples were kept at 2 and 12 respectively. A cell containing the symbol ↑ indicates that that cell contains the same hyperparameter value present in the cell directly above it.

| Dataset | Batch Size $S$ | Encoder epochs | Encoder LR $LR_1$ | BERT seq. len $L_{max}$ |
|---|---|---|---|---|
| LF-WikiSeeAlsoTitles-330K | 1024 | 300 | 0.0002 | 32 |
| LF-WikiTitles-500K | ↑ | ↑ | ↑ | ↑ |
| LF-AmazonTitles-1.3M | ↑ | ↑ | ↑ | ↑ |
| LF-WikiSeeAlso-320K | ↑ | ↑ | ↑ | 128 |
| LF-Wikipedia-500K | ↑ | ↑ | ↑ | ↑ |

# D EVALUATION METRICS

Performance has been evaluated using propensity scored precision@$k$ and nDCG@$k$, which are unbiased and more suitable metric in the extreme multi-labels setting (Jain et al., 2016; Babbar & Schölkopf, 2019; Prabhu et al., 2018a;b). The propensity model and values available on The Extreme Classification Repository (Bhatia et al., 2016) were used. Performance has also been evaluated using vanilla precision@$k$ and nDCG@$k$ (with $k = 1, 3$ and 5) for extreme classification.

Let $\hat{\mathbf{y}} \in \mathbb{R}^L$ denote the predicted score vector and $\mathbf{y} \in \{0, 1\}^L$ denote the ground truth vector (with $\{0, 1\}$ entries this time instead of $\pm 1$ entries, for sake of convenience). The notation $rank_k(\hat{\mathbf{y}}) \subset [L]$ denotes the set of $k$ labels with highest scores in the prediction score vector $\hat{\mathbf{y}}$ and $\|\mathbf{y}\|_1$ denotes the number of relevant labels in the ground truth vector. Then we have:

$$P@k = \frac{1}{k} \sum_{l \in rank_k(\hat{\mathbf{y}})} y_l$$

$$PSP@k = \frac{1}{k} \sum_{l \in rank_k(\hat{\mathbf{y}})} \frac{y_l}{p_l}$$

$$DCG@k = \frac{1}{k} \sum_{l \in rank_k(\hat{\mathbf{y}})} \frac{y_l}{\log(l+1)}$$

$$PSDCG@k = \frac{1}{k} \sum_{l \in rank_k(\hat{\mathbf{y}})} \frac{y_l}{p_l \log(l+1)}$$

$$nDCG@k = \frac{DCG@k}{\sum_{l=1}^{\min(k, \|\mathbf{y}\|_0)} \frac{1}{\log(l+1)}}$$

$$PSnDCG@k = \frac{PSDCG@k}{\sum_{l=1}^{k} \frac{1}{\log l+1}}$$

$$FN@k = 1 - \frac{\sum_{l \in rank_k(\hat{\mathbf{y}})} y_l}{\|\mathbf{y}\|_1}$$

Here, $p_l$ is propensity score of the label $l$ calculated as described in Jain et al. (2016).

## D.1 LABEL QUANTILE CREATION

For Figure 4 and Table 8, labels were divided into 5 equi-voluminous quantiles. To each label $l \in [L]$, a popularity score $V_l = |i : y_{il} = +2|$ was assigned by counting number of training datapoints tagged with that label. The total volume of all labels was computed as $V_{\text{tot}} \stackrel{\text{def}}{=} \sum_{l \in [L]} V_l$. Labels were arranged in decreasing order of their popularity score $V_l$. 5 label quantiles were then created so that the volume of labels in each bin is roughly $\approx V_{\text{tot}}/5$. Thus, labels were collected in the first bin in decreasing order of popularity till the total volume of labels in that bin exceeded $V_{\text{tot}}/5$ at which point the first bin was complete and the second bin was created by selecting remaining labels in decreasing order or popularity till the total volume of labels in the second bin exceeded $V_{\text{tot}}/5$ and so on. For example, for the LF-WikiTitles-500K dataset, the five bins were found to contain approximately $1K, 9K, 30K, 84K, 375K$ labels respectively. Note that the first bin contains very few $\approx 1K$ labels since these are head labels and a small number of them quickly racked up a total volume of $\approx V_{\text{tot}}/5$ whereas the last quantile contains more than $100\times$ more labels at around $375K$ labels since these are tail labels and so a lot more of them are needed to add up to a total volume of $\approx V_{\text{tot}}/5$.

# E  THEORETICAL ANALYSIS

We first recall the notation, then specify Theorem 1 formally, prove the result, and finally extend the result to show that even multiple GCN layers can be approximated using non-GCN networks.

Let $X \in \mathbb{R}^{N \times D} = [\mathbf{x}_1, \ldots, \mathbf{x}_N]^\top$ be the initial embeddings of the $N$ data points over which a graph with adjacency matrix $A \in [0, 1]^{N \times N}$ is present. A typical convolution layer in a GCN can be represented as $\phi(AXW) \in \mathbb{R}^{N \times D}$ where $W \in \mathbb{R}^{D \times D}$ is a transformation matrix and $\phi : \mathbb{R} \to \mathbb{R}$ is some activation function applied coordinate-wise.

(**?**)

**Theorem 2.** *lem:approx[Formal Restatement] Suppose the activation function used in the GCN layer $\phi$ is $\beta$-Lipschitz, i.e., $|\phi(u) - \phi(v)| \leq \beta \cdot |u - v|$ for all $u, v \in \mathbb{R}$. Also suppose there exists a non-GCN (e.g. feedforward, transformer etc.) network $\mathcal{F} : \mathcal{X} \to S^{P-1}$ where $S^{P-1}$ is the the unit sphere in say, $P$ dimensions, that effectively predicts edges in the metadata graph. Specifically, let $\hat{A} = [\hat{a}_{ij}] \in [0, 1]^{N \times N}$ with $\hat{a}_{ij} \overset{\text{def}}{=} (1 + \mathcal{F}(\mathbf{x}_i)^\top \mathcal{F}(\mathbf{x}_j))/2$ be the approximated adjacency matrix. Then for any transformation matrix $W$ utilized by the GCN, there exists exists a non-GCN network $\mathcal{H} : \mathcal{X} \to \mathbb{R}^D$ that well-approximates the embeddings of the GCN layer as well. Specifically, if we abuse notation to let $\mathcal{H}(X) \overset{\text{def}}{=} [\mathcal{H}(\mathbf{x}_1), \ldots, \mathcal{H}(\mathbf{x}_N)]^\top \in \mathbb{R}^{N \times D}$, then we have*

$$\frac{1}{\sqrt{N}} \left\| \phi(AXW) - \mathcal{H}(X) \right\|_F \leq \beta R \cdot \|W\|_2 \cdot \left\| \hat{A} - A \right\|_F,$$

*where $R = \max_{i \in [N]} \|\mathbf{x}_i\|_2$ and $\|W\|_2$ denotes the spectral norm of the matrix $W$.*

Theorem 1 effectively assures us that as $\hat{A} \to A$, we have $\mathcal{H}(X) \to \phi(AXW)$ as well, i.e., the augmented embeddings obtained using the GCN layer can be well-approximated by those offered by the non-GCN network $\mathcal{H}$ if there exists a way to predict the adjacency matrix accurately.

## E.1  PROOF FOR A SINGLE-LAYER GCN

*Proof of Theorem 1.* Consider the network

$$\mathcal{H} : \mathbf{x} \mapsto \phi(T\mathcal{F}(\mathbf{x}) + \mathbf{c}) \in \mathbb{R}^D,$$

where $T \in \mathbb{R}^{D \times P}, \mathbf{c} \in \mathbb{R}^D$ defined as

$$T \overset{\text{def}}{=} \frac{1}{2} \cdot W^\top \left( \sum_{j \in [N]} \mathbf{x}_j \mathcal{F}(\mathbf{x}_j)^\top \right) \in \mathbb{R}^{D \times P}$$

$$\mathbf{c} \overset{\text{def}}{=} \frac{1}{2} \cdot W^\top \left( \sum_{j \in [N]} \mathbf{x}_j \right) \in \mathbb{R}^D$$

Note that $\mathcal{H}$ is a non-GCN network since it merely places a fully connected layer $T$ and a bias term $\mathbf{c}$ on top of a non-GCN network $\mathcal{F}$. Recall that $\mathcal{F} : \mathcal{X} \to S^{P-1}$ and $W \in \mathbb{R}^{D \times D}$ so the dimensionality of $T, \mathbf{c}$ do make sense. Note that the values of the fully connected layer $T$ and the bias term $\mathbf{c}$ depend on the transformation matrix $W$ used by the GCN which implies that for every choice of $W$ made by the GCN layer, there exists a choice of $T, \mathbf{c}$ for the non-GCN network as well.

To prove the result, note that the $i^{\text{th}}$ row of $\phi(AXW)$ can be written as

$$\phi \left( W^\top \left( \sum_{j \in [N]} a_{ij} \mathbf{x}_j \right) \right)$$

whereas the $i^{\text{th}}$ row of $\mathcal{H}(X)$ can be written as

$$\phi\left(T\mathcal{F}(\mathbf{x}_i) + \mathbf{c}\right) = \phi \left( \frac{1}{2} \cdot W^\top \left( \sum_{j \in [N]} \mathbf{x}_j \mathcal{F}(\mathbf{x}_j)^\top \right) \mathcal{F}(\mathbf{x}_i) + \frac{1}{2} \cdot W^\top \left( \sum_{j \in [N]} \mathbf{x}_j \right) \right)$$

$$= \phi \left( W^\top \left( \sum_{j \in [N]} \frac{1 + \mathcal{F}(\mathbf{x}_j)^\top \mathcal{F}(\mathbf{x}_i)}{2} \mathbf{x}_j \right) \right) = \phi \left( W^\top \left( \sum_{j \in [N]} \hat{a}_{ij} \mathbf{x}_j \right) \right)$$

This gives us

$$\|\phi(AXW) - \mathcal{H}(X)\|_F^2 = \sum_{i \in [N]} \left\| \phi \left( W^\top \left( \sum_{j \in [N]} a_{ij} \mathbf{x}_j \right) \right) - \phi \left( W^\top \left( \sum_{j \in [N]} \hat{a}_{ij} \mathbf{x}_j \right) \right) \right\|_2^2$$

$$\leq \beta^2 \cdot \sum_{i \in [N]} \left\| W^\top \left( \sum_{j \in [N]} a_{ij} \mathbf{x}_j \right) - W^\top \left( \sum_{j \in [N]} \hat{a}_{ij} \mathbf{x}_j \right) \right\|_2^2$$

$$= \beta^2 \cdot \left\| (A - \hat{A}) XW \right\|_F^2 \leq \beta^2 \cdot \|XW\|_2^2 \cdot \left\| A - \hat{A} \right\|_F^2$$

where the second step follows since $\phi$ is applied coordinate-wise and is an $L$-Lipschitz function and the last step follows from standard linear algebraic inequalities. We finish the proof by noticing that the spectral norm is submultiplicative and $\|X\|_2 \leq R\sqrt{N}$. $\qquad\square$

### E.2 EXTENSION TO GCNS WITH MULTIPLE LAYERS

We note that this result can be extended to multiple layers. For example, suppose we wish to utilize two graph convolution layers i.e.

$$\phi(A\phi(AXW)\tilde{W}),$$

where $\tilde{W} \in \mathbb{R}^{D \times D}$ is the transformation matrix for the second GCN layer. The proof technique presented above can be extended to show that the following non-GCN network would approximate the above two-layer GCN.

$$\mathcal{K} : \mathbf{x} \mapsto \phi(\tilde{T}\mathcal{F}(\mathbf{x}) + \tilde{\mathbf{c}}) \in \mathbb{R}^D$$

where

$$\tilde{T} \stackrel{\text{def}}{=} \frac{1}{2} \cdot \tilde{W}^\top \left( \sum_{j \in [N]} \mathcal{H}(\mathbf{x}_j) \mathcal{F}(\mathbf{x}_j)^\top \right) \in \mathbb{R}^{D \times P}$$

$$\tilde{\mathbf{c}} \stackrel{\text{def}}{=} \frac{1}{2} \cdot \tilde{W}^\top \left( \sum_{j \in [N]} \mathcal{H}(\mathbf{x}_j) \right) \in \mathbb{R}^D$$

where $\mathcal{H}$ is the non-GCN network explicated in the proof of Theorem 1. This technique can be extended to cases with more than 2 GCN layers as well.

## F ETHICAL CONSIDERATIONS

Our usage of data and terms of providing service to people around the world has been approved by our legal and ethical boards. In terms of social relevance, our research is helping millions of people find the goods and services that they are looking for online with increased efficiency and a significantly improved user experience. This facilitates purchase and delivery without any physical contact which is important given today's social constraints. Furthermore, our research is increasing the revenue of many small and medium businesses including mom and pop stores while also helping them grow their market and reduce the cost of reaching new customers.

