# OpenReview forum: "Graph Regularized Encoder Training for Extreme Classification"
_ICLR.cc/2025/Conference — ICLR 2025 Conference Withdrawn Submission_

### Official Review · Reviewer_Vyqe · 2024-10-30

**Soundness:** 3
**Presentation:** 1
**Contribution:** 2
**Rating:** 6
**Confidence:** 3

**Summary:**

The paper introduces RAMEN, a novel approach designed to enhance extreme classification (XC) tasks by leveraging graph metadata to regularize encoder training. RAMEN effectively addresses the limitations of GNN-based methods, such as computational inefficiency and vulnerability to noisy graph data. By utilizing graph metadata as a regularizer rather than integrating GCN layers directly, RAMEN achieves significant performance improvements, up to 15% higher accuracy on benchmark datasets, while maintaining scalability and minimizing inference costs. The method demonstrates robust performance across various datasets, including proprietary recommendation systems, underscoring its applicability and effectiveness in handling both dense and sparse label scenarios.

**Strengths:**

1. **Innovative Methodology:** The paper proposes a novel graph-regularized training framework, RAMEN, specifically designed for extreme classification tasks. This method effectively mitigates the limitations of traditional GCN-based approaches, such as high computational overhead and sensitivity to noisy graph structures. By regularizing the encoder with graph metadata during training, RAMEN enhances model robustness and efficiency, enabling scalability to datasets with millions of labels without increasing inference costs.

2. **Comprehensive Experimental Validation:** The authors conduct extensive experiments across multiple benchmark datasets, including LF-WikiSeeAlsoTitles-320K, LF-WikiTitles-500K, and LF-AmazonTitles-1.3M, as well as a proprietary dataset (G-EPM-1M). RAMEN consistently outperforms baseline methods, including both text-based and graph-based approaches, by up to 15% in precision metrics. Additionally, ablation studies demonstrate the effectiveness of key components such as bandit learning for regularization constants and graph pruning for handling noisy metadata. The real-world application in sponsored search further validates RAMEN's practical utility, showcasing significant improvements in impression and click yields during A/B testing.

**Weaknesses:**

1. **Related Work Presentation:** The related work section primarily lists references without adequate categorization or in-depth analysis. This approach limits the ability to understand how RAMEN differentiates itself from existing methodologies and fails to provide insightful connections between prior studies and the proposed approach. A more structured review, categorizing related works based on their methodologies and highlighting comparative strengths and weaknesses, would enhance the comprehensiveness and clarity of the literature context.

2. **Clarity and Formal Description:** The presentation of certain concepts, particularly the comparison between GCN-based extreme classification methods and RAMEN, lacks clarity. The paper would benefit from a more precise formal description of GCN-based extreme classification methods, including explicit definitions and step-by-step explanations. Additionally, outlining the limitations of these methods in a clear, point-wise manner would aid readers in better understanding the motivations behind RAMEN and the specific challenges it addresses.

**Questions:**

1. **Concrete Examples of Graph Metadata:**
   Can the authors provide specific examples of the types of graph metadata utilized in their experiments? For instance, what defines an anchor set in the different datasets (e.g., LF-WikiSeeAlsoTitles-320K, LF-WikiTitles-500K, LF-AmazonTitles-1.3M), and how are these metadata graphs constructed and integrated within the RAMEN framework?

2. **Robustness to Noisy Graphs:**
   How does RAMEN perform when exposed to explicitly added noisy graph metadata? Specifically, if additional irrelevant or incorrect edges are introduced into the metadata graphs, what impact does this have on RAMEN's accuracy and robustness compared to traditional GCN-based methods? Have the authors conducted any experiments to evaluate RAMEN's resilience to such noise?

3. **Adaptability to Other Domains:**
   The experiments presented focus on datasets derived from Wikipedia and Amazon. How adaptable is RAMEN to other domains that have different types of graph metadata, such as social networks, biomedical data, or recommendation systems in other industries? Have the authors conducted any preliminary tests in these diverse domains, or are there plans to evaluate RAMEN's performance in such settings?

---

> ### Author Response · Authors · 2024-11-22
>
> > Robustness to Noisy Graphs:
>
> Thank you raising the question. Noisy graph in metadata can impact training and inference. RAMEN analyse the noisy in both steps.
> - In training Table 5(No Pruning) in Ablation shows that when noisy nodes are not pruned dynamically during training, RAMEN observes the loss on 5 points in P@1. Show casing metadata graph have noise and RAMEN can robustly handle this noise which lead to state-of-the-art results
> - In inference, RAMEN is does not use any graph information during inference. However, GCN methods like OAK, induced graph in over test points to make predictions. Table 2 shows that, inducing graph leads to 2 points loss in P@1. However, if we have oracle graph information (i.e. graph with no noise), OAK can outperform RAMEN. It should be noted that oracle graph is never available for a novel test point.
>
> > Adaptability to Other Domains:
>
> We thank the reviewer for asking this question. RAMEN was tested in the following domains
> - Wikipedia
> - Amazon
> - Online recommendation scenarios.
>
> RAMEN was deployed on a popular search engine in 160+ countries with hundreds of millions of labels where RAMEN showed 2.8% gains in ads impressions. We encourage reviewer to refer to line 492-502.

---

### Official Review · Reviewer_F49Y · 2024-10-31

**Soundness:** 3
**Presentation:** 2
**Contribution:** 2
**Rating:** 3
**Confidence:** 4

**Summary:**

This paper presents a novel approach, RAMEN, for extreme classification (XC) that effectively utilizes graph metadata to regularize encoder training. The method shows significant performance improvements over state-of-the-art techniques, with up to 15% higher prediction accuracies on benchmark datasets. The theoretical foundation provided by Theorem 1 and its extensions is a strong aspect of the paper, validating the approach. The extensive experimental results, including ablation studies, further demonstrate the effectiveness of RAMEN. However, the paper could benefit from addressing some weaknesses.

**Strengths:**

1. The paper presents a novel method, RAMEN, for leveraging graph metadata in extreme classification (XC) settings. It offers an alternative to Graph Convolutional Networks (GCNs) by using graph data to regularize encoder training.
2. The paper conducts extensive experiments on multiple benchmark datasets and a proprietary dataset.

**Weaknesses:**

1. The process of tuning the regularization constants using the bandit learning approach may be hard to follow for some readers. The initialization of the constants and the perturbation process every 30 iterations, along with the update rule using the estimated gradient, involves multiple steps and concepts.
2. The Related Work of this paper lists many references related to XC methods without providing in-depth analysis of how the proposed method in this paper (RAMEN) overcomes the limitations of these existing methods in a more detailed and unique way. While it gives an overview of the field's development, it doesn't clearly distinguish the novelty and superiority of the new approach in a concise manner.
3. The method and its associated theoretical concepts may be complex for some readers to understand. The notation and mathematical derivations, especially in the sections related to Theorem 1 and its proof, could be a barrier to a broader audience. Simplifying or providing more intuitive explanations could enhance the paper's accessibility

**Questions:**

Terms like "in-batch negatives anchors" and the specific way the loss function  and regularization functions interact to encourage the encoder to embed data points and labels in a certain way could be better explained.

---

> ### Author Response · Authors · 2024-11-22
>
> We understand the reviewer's perspective of improving writing quality and some clarification questions, we would reviewer to re-consider their score as it doesn't reflect, he novelty of our method.
>
> > Terms like "in-batch negatives anchors" and the specific way the loss function and regularization functions interact to encourage the encoder to embed data points and labels in a certain way could be better explained.
>
> We will improve the clarity of paper writing. For the rebuttal here is short description of how we are creating in batch negative anchors. Say we have a mini-batch of 1024 document-label pair, for each document and label in the mini-batch we sample one pair of anchor set, that adds to 1024 document-anchor set pair and 1024 label-anchor set pair. We compute loss function over similarity score matrix (1024x1024) of document-label pairs, similarly for score matrix (1024x1024) of document-anchor pair and label-anchor pair During training. In this score matrix, diagonals are positive pair and off diagonals are negative pairs.
>
> > process of tuning the regularization constants using the bandit learning approach may be hard to follow for some readers
>
> We will improve the writing. For rebuttal, here is the intuition of using bandits. In a mini batch during training, bandits are responsible to controlling the influence of gradients from multiple loss functions such that overall recall of learned model for predicting relevant labels for a query is maximised.
>
> > many references related to XC methods without providing in-depth analysis of how RAMEN overcomes the limitations
>
> Each method in XC has contributed significantly to advancement of the field. While we cannot cover all methods in XC, we have covered important baselines of for RAMEN in line 141-165. Namely section **Label metadata in XC** and **Graph Neural Networks in Related Areas**.
>
> > method and its associated theoretical concepts may be complex for some readers to understand
>
> As stated before, we will improve the writing quality

---

### Official Review · Reviewer_Gd2A · 2024-11-03

**Soundness:** 1
**Presentation:** 3
**Contribution:** 1
**Rating:** 3
**Confidence:** 4

**Summary:**

This paper presents RAMEN, a method for extreme classification with graph metadata, primarily focusing on ranking and recommendation scenarios. While the authors address the challenge of handling new users and items without graph edges, **they notably overlook the well-established concept of "cold-start" recommendation - a significant oversight in the recommendation systems literature**. The absence of comparisons with existing cold-start solutions and related work makes it difficult to properly evaluate the paper's true contributions to the field. Additionally, the experimental validation is limited by the relatively small dataset size of 1.3 million samples, which fails to demonstrate the method's efficacy in large-scale, real-world applications. Without addressing these fundamental concerns - namely, the missing cold-start comparisons and limited dataset scale - the paper's claims of superiority over existing methods remain inadequately supported. These limitations significantly impact the paper's potential contribution to the field and its suitability for publication at a prestigious venue like ICLR 2025.

**Strengths:**

1. Graph Regularization Approach: Instead of using complex GCN, the paper proposes a simpler yet effective approach of using graph data for regularization. The method is theoretically supported and mathematically sound.
2. Inference Efficiency: RAMEN achieves faster prediction by eliminating graph traversal during test time, while maintaining accuracy. The two-stage comparison with GCN methods is clearly visualized and demonstrates practical advantages.
3. Multi-Graph Flexibility: The system can handle multiple graph types simultaneously and integrates easily with existing systems. This versatility is well-demonstrated through experimental results and architectural design

**Weaknesses:**

1. Critical Cold-Start Problem Oversight:The paper fundamentally fails to acknowledge or address the cold-start problem, despite directly tackling scenarios where new items have no graph connections. This is not merely a terminology issue - it's a severe oversight that ignores decades of research in recommendation systems. The absence of comparisons with established cold-start solutions makes it impossible to validate the authors' claims of superiority.
2. Inadequate Dataset Scale: Using only a 1.3M dataset in an era where real-world recommendation systems handle hundreds of millions of users/items is a critical limitation. This small-scale evaluation raises serious doubts about RAMEN's practical applicability and scalability claims. The paper's claims about handling "extreme" classification become questionable when tested on such modest datasets.
Missing Technical Validation
3. The paper lacks crucial technical validations, including ablation studies, complexity analysis, and detailed parameter sensitivity tests. The implementation details are vague, making reproduction difficult. Without these essential technical components, the paper's theoretical contributions remain unsubstantiated by empirical evidence, particularly for real-world deployment scenarios.

**Questions:**

Shown in weakness

---

> ### Author Response · Authors · 2024-11-22
>
> Thanks for your review. We believe our paper already addressed two of your three concerns; we discuss the third one here, and can add it to the paper.
> > overlook the well-established concept of "cold-start" recommendation - a significant oversight
>
> Cold start recommendation [1] is defined in two scenarios,
> - New user on platform: This is analogous to unseen test points.
> - New items to recommend: This is analogous to predicting for unseen labels, also known as zero shot.
> In extreme classification, for evaluation all test are never seen during training therefore making it cold start with respect to new user. In addition to this, we have compared RAMEN with state-of-the-art XC method on zero shot scenario in Table 12. In both scenarios RAMEN regularized encoder can outperform  baselines by 5 and 2 points repectively.
>
> * [1] [Cold Start Thread Recommendation as Extreme Multi-label Classification](https://dl.acm.org/doi/10.1145/3184558.3191659)
>
> > relatively small dataset size of 1.3 million samples
>
> In the interest of reporting performance on well-known public data sets, we chose data sets with up to 1.3 million instances. However, we request the reader to inspect Table 11, described around line L492-L502. Here we used proprietary data from a live search engine, comprising **540 million training documents and 360 million labels**.
>
> > lacks … ablation studies, complexity analysis, and detailed parameter sensitivity tests
>
> **Ablation:** Table 5 presents various ablations, e.g., changing the base method to featurize label meta-data, excluding the bandits or pruning strategies, removing the graph information on the document or label side, etc. Detailed discussions on the ablations are seen in and around page 9, as well as lines L503-L527. Please inspect these parts and let us know in case you have further questions.
>
> **Complexity:** On line L283 we make clear that RAMEN incurs no additional inference time over the closest baselines; all the work is relegated to training time. Further, Table 3 shows that, compared to ANCE (which is the fastest to train), RAMEN is only 10% slower during training.
>
> We further add that, in the context of irregular data artifacts like instance and label metadata graphs, simple closed-form expressions for complexity are not usually meaningful (because different instances and labels are connected to such diverse graph neighborhoods).
>
> **Parameter sensitivity:** Appendix B and Table 16 give sufficient details about parameters for reproducing our results. In addition, we intend to make our code and non-proprietary data sets public.

---

### Official Review · Reviewer_To6S · 2024-11-03

**Soundness:** 3
**Presentation:** 1
**Contribution:** 2
**Rating:** 5
**Confidence:** 2

**Summary:**

The goal of the RAMEN model is to leverage graph metadata during training without introducing extra inference time costs. The graph data is used to regularize the encoder $\mathcal{E}$ by embedding points close to the representations of correct labels and distant from the representations of incorrect labels. The authors propose a contrastive regularization loss function that utilizes this graph metadata to improve classification accuracy, especially for rare labels.

**Strengths:**

**Evaluation and Performance**: The experimental results are promising. RAMEN demonstrates  improvements in multi-labels performance over multiple baseline methods.

**Effective method**: Table 3 shows that RAMEN show impressive gain in term of inference and training.

**Weaknesses:**

**Presentation**: The model figure could be more detailed. Currently, it repeats parts of the abstract verbatim, and a more informative figure would have helped clarify the steps, especially how the graph metadata regularizes the encoder through the loss function. Similarly, an introductory figure outlining the overall method would improve readability. Table 4 also needs cropping; important parts are not visible, making it hard to interpret.

**Clarity of Contributions:** The contributions of the paper are difficult to follow. From what I understand, there is a contrastive loss that uses the graph metadata to define positive and negative examples, but this is not explained in sufficient detail, making it hard to fully grasp the model’s novelty and mechanisms. Also the training with

**Typos**: There are some typo errors. For example, Line 1035 has a "(?)" and Line 1036 seems to have an incomplete reference "Lem[...]".

**Questions:**

1. In the metadata graph regularization, is the graph primarily used to define positive and negative links for the contrastive loss? Please clarify if the positive and negative examples come solely from this graph.

2. Theorem 1: If I understand correctly, the theorem suggests that if an adjacency matrix can be approximated by a non-GCN (e.g., feedforward network or transformer), it implies the existence of another non-GCN that can approximate $\phi(AXW)$ However, is this new network permutation invariant? The usual motivation for using GCNs is precisely their permutation invariance across the matrix $A$. Therefore, I am not fully convinced by this argument based on my current understanding of the theorem.

---

> ### Author Response · Authors · 2024-11-22
>
> Thanks for the generally positive comments on soundness and contribution, and also for pointing out many limitations in the presentation. Here we attempt to clarify some of your doubts caused by the presentation. We have also updated the manuscript with some of these clarifications.
>
> > In the metadata graph regularization, is the graph primarily used to define positive and negative links for the contrastive loss? Please clarify if the positive and negative examples come solely from this graph.
>
> In graph regularization, graphs are used to identify positive links between, document/label to their corresponding anchor set. Say we have a mini-batch of 1024 document-label pair, for each document and label in the mini-batch we sample one pair of anchor set, that adds to 1024 document-anchor set pair and 1024 label-anchor set pair. During training, negatives for a document-label batch comes from the labels sampled from the minibatch. Similary for document-anchor or label-anchor set negatives comes from anchors sampled from the mini-batch.
>
> > If I understand correctly, the theorem suggests that if an adjacency matrix can be approximated by a non-GCN (e.g., feedforward network or transformer), it implies the existence of another non-GCN that can approximate … However, is this new network permutation invariant? The usual motivation for using GCNs is precisely their permutation invariance across the matrix. Therefore, I am not fully convinced by this argument based on my current understanding of the theorem.
>
> Thank you for asking this question, yes the new network is permutation invariant, theorem does not assume any order of linked graph metadata, and permuting the sequence of metadata does not change the final outcome.
>
> > model figure could be more detailed … more informative figure would have helped clarify the steps, especially how the graph metadata regularizes the encoder through the loss function.
> > Similarly, an introductory figure outlining the overall method would improve readability.
>
> We will update the figure with better details and clairty.
>
> > Table 4 also needs cropping; important parts are not visible, making it hard to interpret.
>
> Our apologies. "Table" 4 is actually a bar-chart, and will be renamed as a figure. But all necessary parts of the bar-chart are visible to us on overleaf with standard, unmodified style files.
>
> > Line 1035 has a "(?)" and Line 1036 seems to have an incomplete reference "Lem[...]".
>
> Thanks for pointing out the LaTeX error, it will be fixed in the revised edition.

---

### Note · Authors · 2024-11-22

I have read and agree with the venue's withdrawal policy on behalf of myself and my co-authors.